# Advances in Research on Diabetes by Human Nutriomics

**DOI:** 10.3390/ijms20215375

**Published:** 2019-10-29

**Authors:** Xinmin Ren, Xiangdong Li

**Affiliations:** 1Beijing Advanced Innovation Center for Food Nutrition and Human Health, China Agricultural University, Beijing 100193, China; R15910206682@163.com; 2State Key Laboratory of Agrobiotechnology, College of Biological Sciences, China Agricultural University, Beijing 100193, China

**Keywords:** diabetes mellitus (DM), nutrigenomics, nutritional-metabolomics, foodomics, molecular biomarkers

## Abstract

The incidence and prevalence of diabetes mellitus (DM) have increased rapidly worldwide over the last two decades. Because the pathogenic factors of DM are heterogeneous, determining clinically effective treatments for DM patients is difficult. Applying various nutrient analyses has yielded new insight and potential treatments for DM patients. In this review, we summarized the omics analysis methods, including nutrigenomics, nutritional-metabolomics, and foodomics. The list of the new targets of SNPs, genes, proteins, and gut microbiota associated with DM has been obtained by the analysis of nutrigenomics and microbiomics within last few years, which provides a reference for the diagnosis of DM. The use of nutrient metabolomics analysis can obtain new targets of amino acids, lipids, and metal elements, which provides a reference for the treatment of DM. Foodomics analysis can provide targeted dietary strategies for DM patients. This review summarizes the DM-associated molecular biomarkers in current applied omics analyses and may provide guidance for diagnosing and treating DM.

## 1. Background

Diabetes mellitus (DM) is a metabolic disorder characterized by prolonged periods of hyperglycemia [1], which includes frequent urination, thirst, and hunger [2]. The three main types of DM are as follows [3]: Type 1 diabetes (T1D), also known as “insulin-dependent diabetes mellitus” (IDDM) or “juvenile diabetes”, is caused by the inability of the pancreas to produce sufficient insulin due to beta cell loss. Type 2 diabetes (T2D), also known as “non-insulin-dependent diabetes mellitus” (NIDDM) or “adult-type diabetes”, begins with insulin resistance, and its progression may involve a lack of insulin. Gestational diabetes mellitus (GDM) refers to hyperglycemia in pregnant women with no previous history of DM. Failure to treat any of these types in time causes many complications [4]. Acute complications include diabetic ketoacidosis, hyperosmolar hyperglycemia, and even death. Chronic complications include cardiovascular disease (CVD), stroke, chronic kidney disease, foot ulcers, and eye damage.

According to the International Diabetes Federation (IDF), 425 million DM patients existed worldwide in 2017, and this number is estimated to increase to 629 million by 2045 [5]. In 2017, DM patients in China ranked first worldwide, which is a value of 114,294.8 (https://www.idf.org/). However, because DM is extremely heterogeneous, individualized treatments are required [6]. Comprehensive knowledge of the pathogenic factors of DM is becoming urgently necessary. Most studies have collected clinical cases, summarizing and suggesting factors that may promote DM occurrence. Since the application of multi-data nutritional-omics analysis methods in DM research, obtaining potential molecular markers of pathogenicity is highly straightforward and provides a shortcut for further research, clinical diagnosis, and treatment.

Human nutriomics is a new discipline formed from the combination of nutritional food science and omics in the post-genomic era, and mainly involves studies of dietary intervention methods and nutritional health measures at both the molecular and population levels to achieve personalized nutrition [7]. Human nutriomics typically include protein-based human nutrigenomics, metabolic component-based human nutritional-metabolomics, microbial-based human nutritional systems biology, system-based food-omics, and systems biology. This review analyzed nutriomics over the years, described the research progress and molecular mechanisms, and provides a reference for subsequent DM research.

## 2. Human Nutrigenomics and DM

Human nutrigenomics, a new field in the study of nutrition proposed in 2000 [8], is the study of the molecular biological processes and effects of nutrients and food chemicals in humans [9]. Studying the transcription, translation, and expression of human genes and metabolic mechanisms enables establishing dietary recommendations with high predictive value to prevent disease, reduce the risk of unpredictable consequences, and control chronic diseases [10]. Protein-based human nutrigenomics has three research directions [11]: nutritional genomics, transcriptomics, and proteomics.

### 2.1. Advances in DM Research via Nutritional Genomics

Genomics was first proposed by Roderick et al. in 1986, using DNA mapping, sequencing, and bioinformatics techniques to analyze the structure and function of all genomes in living organisms [12]. Nutritional genomic research methods are consistent with functional genomics research [13], mainly including DNA chip technology, biomarkers, and proteomic technologies. Nutritional genomics generally uses genome-wide association studies (GWAS) to analyze the pathogeneses of diseases, such as DM. GWAS involves identifying existing sequence variants, which include single nucleotide polymorphisms (SNPs) in human genome-wide applications, from which disease-associated SNPs are screened [14]. Studies have shown that using binary logistic regression analysis to analyze diabetes can effectively determine various DNA sequence variations [15,16].

GWAS have undergone several updates and upgrades. The latest T2D GWAS showed that common variations explain approximately 20% of the overall T2D risk, which is equivalent to at least half of the overall heritability [17,18]. An analysis of the common SNPs in T2D patients and matched controls identified three T2D-associated loci in the noncoding regions near *CDKN2A* and *CDKN2B*, introns of *IGF2BP2* and *CDKAL1*, as well as replication associations near *HHEX* and *SLC30A8*. T2D-related loci were also identified and confirmed in noncoding regions by analyzing common SNPs in T2D patients and matched controls [19]. Jeong et al. (2019) used DNA microarray to analyze cases of diabetic nephropathy (DN) and control cases and found that rs3765156 in PIK3C2B was significantly associated with DN [20]. Using epigenome analysis of primary TH1 and TREG cells isolated from healthy and T1D subjects, Gao et al. (2019) identified four SNPs (rs1077211, rs1077212, rs3176792, and rs883868) that could alter enhancer (H3K4me1 and H3K27ac) activities [21]. After finding a DM-related candidate gene through GWAS, experimental verification (i.e., cell and animal experiments) is essential.

### 2.2. Advances in DM Research via Transcriptomics

Transcriptomics refers to the sum of all gene transcripts of an organism or cell under certain conditions, which contain the protein-encoding RNA required by the cell for a specific time and environment and a collection of RNA molecules derived from the expression-regulating gene [22]. Transcriptomics is based on sequencing technology development, which includes expression sequence tags technology (EST), serial analysis of gene expression (SAGE), massive parallel signature sequencing (MPSS), and RNA-seq. Many updated databases can be used for research, such as GEO, SRA, DDBJ, NONCODE, MIRBase, ERA, and DRA.

Establishment of a transcriptome database has facilitated DM-related research. Wieczorek et al. (2019) examined transcriptomic data from the salivary gland tissues of T1D patients and found inhibited ectopic lymphoid structures and Sjögren’s syndrome by blocking the CD40–CD154 pathway interaction [23]. Using microarray-based DM and normal glucose tolerance, transcriptional profiles of subcutaneous adipose tissue have been generated via studies from Asia and India. Differentially expressed genes can be analyzed using weighted gene coexpression network analysis to clinically diagnose DM [24]. Researchers used a whole transcriptome and small RNA analysis to optimize clinical tissue samples from DM patients with cardiovascular disease and successfully completed the whole transcriptome from human left ventricular tissue [25]. Fang et al. (2019) developed RePACT, a sensitive single-cell analysis algorithm, to discover the previously unrecognized role of the cohesin-loading complex and the NuA4/Tip60 histone acetyltransferase complex in regulating insulin transcription and release [26]. Hong et al. (2019) identified a number of proangiogenic genes in transcriptomic features of glomerular endothelial cells, including leucine-rich α-2-glycoprotein 1 (LRG1), which was upregulated in DM mouse models [27]. Caberlotto et al. (2019) analyzed transcriptomic data from the brains of postmortem Type 3 Diabetes (T3D) and T2D patients to determine the main role of autophagy in the molecular basis of T3D and T2D [28]. Dusaulcy et al. (2019) analyzed islet cell transcriptome data from control and DM-induced mice, revealing 11–39 differentially expressed genes in the transcriptome of pancreatic alpha cells from obese hyperglycemic mice compared with controls, and identified three new target genes (*Upk3a, Adcy1*, and *Dpp6*) after further analysis [29]. In summary, construction and analysis of a transcriptome database provides a reliable basis for researching DM mechanisms and determining gene functions.

### 2.3. Advances in DM Research via Proteomics

Wilkins and Williams first proposed the concept of proteomics at the First International Proteomics Workshop in Italy in 1994 [30]. Proteomics refers to the science of understanding the laws of life activities from the whole protein level, with the proteome as the research object. Proteomics represents a mature technology in the pharmaceutical industry, primarily for discovering biomarkers and drug targets [31]. Proteomics is divided into expression proteomics, structural proteomics, and functional proteomics, and, mainly through two-dimensional gel electrophoresis (2-DE), mass spectrometry (MS), and other methods, is used to study protein function and disease mechanisms.

Related nutrient proteins can be controlled from a dietary perspective to achieve early prevention and early treatment using proteomics methods to study nutritional diseases. Sramkova et al. (2019) identified apolipoprotein M (apoM) by transcriptome and proteomic analysis of conditioned media from human adipose tissue (AT)-isolated adipocytes and stromal cells, in which the expression level is lower in subjects with metabolic syndrome and T2D and may be associated with insulin sensitivity [32]. Abdulwahab et al. (2019) collected sera from healthy people and T2D patients for proteomic mass spectrometry and found that 62 proteins were differentially expressed in T2D, which were functionally grouped into 16 proteins, including heparin cofactor 2, Ig α-1 chain C region, and zinc-α-2-glycoprotein, the largest of which was an immune-related protein [33]. Muralidharan et al. (2019) used borate affinity chromatography to isolate glycated erythrocyte proteomes without hemoglobin from controls and DM samples, and proteomic analysis, using the nanoLC/ESI-MS proteomics platform, to identify site-specific glycation of the red blood cell proteome with different glycemic indices in DM patients [34]. Ricci et al. (2019) evaluated peptide biomarkers, using capillary electrophoresis and mass spectrometry (CE-MS), and demonstrated that the urinary proteome of pediatric renal cysts and diabetes syndrome (RCAD) patients differs from that of autosomal dominant polycystic kidney disease (PKD1, PKD2), congenital nephrotic syndrome (NPHS1, NPHS2, NPHS4, and NPHS9), and chronic kidney DM conditions, suggesting differences between the pathophysiologies of these disorders [35]. Malipatil et al. (2019) used sequential window acquisition of all theoretical fragment ion spectroscopy (SWATH) MS to study the biodeterminants associated with the response to diet and weight loss programs in impaired glucose regulation populations. These authors successfully differentiated individuals who may lose weight from those who may experience increased insulin sensitivity. For example, as insulin sensitivity improves, hemoglobin A1c (HbA1C) levels decrease with weight loss [36]. Rao et al. (2007) used the differential in-gel electrophoresis method to compare protein groups in DM and normal human urine. Seven upregulated proteins (α_1B_-glycoprotein, zinc-α_2_-glycoprotein, α_2_-*HS*-glycoprotein, VDBP, calgranulin B, A1AT, and hemopexin) and 4 downregulated proteins (prealbumin, α_1_-microglobulin, bikunin, and apoB/A1) in DM patients can be used as additional tests to diagnose DM [37].

The role of proteomics in studying DM complications is important. Sims et al. (2014) used proteomics to screen for proteins specifically expressed in the urine of patients with DM retinopathy and nephropathy to find biomarkers for early diagnosis [38]. Chiang et al. (2012) used 2-DE and MALDI-TPF-MS to analyze protein expression in diabetic retinopathy (DR) patients and identified 11 differentially expressed proteins associated with nutrient transport, microstructural reorganization, angiogenesis, antioxidants, and neuroprotection [39]. Zoccali et al. (2019) focused on diabetic kidney disease (DKD) proteome biomarkers. These authors found that the DKD 273 classifier was a promising biomarker for early identification of nonproteinuric patients at high risk for progressive DKD. Empagliflozin and SGLT2 inhibitors may favorably affect DKD progression in nonalbuminuric diabetic patients [40]. Mirza et al. (2014) summarized the proteomic relationship between T3D and T2D, identifying a single or set of potential blood-based protein biomarkers with high sensitivity and specificity for early diagnosis of AD and T2D [41].

## 3. Human Nutritional Metabolomics and DM

Nutritional metabolomics is a means of studying the relevant issues in the field of nutrition via metabolomic principles and methods [42]. Metabolomics focuses on the metabolic pathways of endogenous small molecule metabolites in organisms, organs, and tissues and their changes, which can reflect the end point of the physiological regulation process in real time. The obtained information is closest to the organismal phenotype or overall condition and is the final expression of the biological phenomenon. Nutritional metabolomics can be divided into metabolomics, lipidomics, and metallomics.

### 3.1. Advances in DM Research via Metabolomics

In 1999, Nicholson et al. (1999) from Imperial College of Science and Technology in the United Kingdom proposed nuclear magnetic resonance-based metabonomics based on long-term graduated body fluids [43]. Fiehn et al. (2002) used gas chromatography to study the plant metabolism network and simultaneously proposed metabolomics [44]. As the research progressed, metabonomics and metabolism blended together and now represent metabolomics [45]. Metabolomics mainly analyzes the association of endogenous small molecular metabolites with physiological and pathological changes under the influence of internal and external factors (such as genetic variation, disease invasion, drug intervention, and environmental changes) by group indicators and is mainly divided into the levels of metabolic target analysis, metabolic profiling, metabolic fingerprinting, metabolomics analysis, and metabolic phenotypic analysis.

Compared with nutritional genomics, transcriptomics, and proteomics, metabolomics has the advantages of obvious changes, fewer species and quantities, specific components, common methods, and lower costs. Vangipurapu et al. (2019) studied metabolomic data on 20 amino acids from 4851 patients with cross-sectional metabolic syndrome and found that the expressions of five amino acids (tyrosine, alanine, isoleucine, aspartic acid, and glutamic acid) were significantly associated with an increased risk of developing T2D [46]. Bernardo–Bermejo et al. (2019) developed a liquid chromatography-mass spectrometry platform for the nontargeted metabolomic analysis of high glucose-induced changes in human proximal tubular H2 cell cultures to study renal proximal tubules in the DM mechanisms in kidney disease progression [47]. Using integrated transcriptomic-metabolic methods, Osataphan et al. (2019) demonstrated that canagliflozin (CANA) regulates key nutrient-sensing pathways, activates 5’AMP-activated protein kinase (AMPK), and inhibits rapamycin (mTOR) independent of insulin or glucagon sensitivity or signaling [48]. Using metabolomics to analyze food nutrients contributes to preventing and treating metabolic diseases through dietary intervention.

### 3.2. Advances in DM Research Using Lipidomics

Lipidomics is the study of lipid extracts to obtain lipid group information that reflects the overall changes in lipids under specific physiological conditions [49]. At present, the many branches of lipidomics include cell lipidomics, computational lipidomics, and neurolipidology. Commonly used techniques related to lipidomics are thin-layer chromatography (TLC), electrospray ionization mass spectrometry (ESI-MS), gas chromatography mass spectrometry (GC-MC), high performance liquid chromatography coupled with time-of-flight mass spectrometry detection (HPLC-TOF/MS), ultra-performance liquid chromatography coupled with quadrupole time-of-flight/ mass spectrometry (UPLC-TOF/MS), matrix-assisted laser desorption ionization time-of-flight mass spectrometry (MALDI-TOF-MS), and shotgun lipidomics.

Lipidomics research has shown that metabolic diseases, such as obesity and DM, are closely related to lipid metabolic disorders [50]. Lipidomics has enabled important advances in detecting metabolic diseases, identifying lipid biomarkers and drug targets, and developing new drugs. He et al. (2019) established a lipid mass spectrum of 29 women with GDM and 33 pregnant women without GDM and found that elevated GPR120 levels were associated with GDM [51]. Lamichhane et al. (2019) compared the cord blood lipids of T1D patients with those of healthy children and found that phospholipids, especially sphingomyelin, were lower in T1D progression [52]. Wang et al. (2019) screened spontaneously obese rhesus monkeys and performed plasma lipidomics analysis on both normal weight and obese monkeys using gas chromatography/mass spectrometry (GC/MS) and ultraperformance liquid chromatography/mass spectrometry (UPLC/MS). These authors found that FFA C16:0 and 16:0-LPA lipids may be potential candidates for diagnosing and studying obesity-related diseases [53]. Zhang et al. (2019) constructed a Paternò–Büchi reaction coupled with liquid chromatography/mass spectrometry (LC/MS), using an online C=C derivatization lipid analysis platform, and found that the C=C isomer can be used to discover lipid biomarkers, which can be used for subsequent predictive DM screening [54]. Pape et al. (2018) explored the role of high-fat-diet-induced hepatic triglycerides in high-fat-diet-induced DM by MS-based lipidomics and found correlations between Per-Arnt-Sim Kinase (PASK) [55]. Lamichhane et al. (2018) reported a longitudinal plasma lipidomics dataset from 40 children who had progressed to T1D and 40 children with single islet autoantibodies without T1D and 40 matched controls. Their data could help other researchers study the age-dependent progression of islet autoimmunity and T1D as well as the age-dependent nature of the general lipidomics spectrum [56]. Razquin et al. (2018) randomly selected 692 participants (639 non-cases and 53 T2D cases) and repeatedly measured 207 plasma lipid metabolites. They used principal component analysis to establish a comprehensive factor for lipid species and evaluated the association between these lipid factors and T2D incidence [57]. Zhai et al. (2018) used lipidomics analysis and found that *Cyclocarya paliurus* may improve diabetic dyslipidemia by reducing accumulation of hepatic lipid droplets and regulating circulatory lipids in diabetic mice via PI3K signaling and MAPK signaling pathways. [58]. Yang et al. (2018) analyzed large datasets generated by metabolomics and lipidomics and revealed the role of metabolites, such as lipids, amino acids, and bile acids, in regulating insulin sensitivity [59].

### 3.3. Advances in DM Research Using Metallomics

Metallomics is the comprehensive study of the distribution, presence, content, structural characteristics, and physiological functions of metal and metalloid elements in biological systems [60]. Metallurgical research includes the distribution and analysis of elements in organs, tissues, body fluids, and cells in organisms; the morphological analysis of elements in biological systems; the structural analysis of metallography; the mechanism of metal group reactions; the action of metalloproteins and metalloenzyme identification; metabolic analysis of biomolecules and metals; multi-element analysis of medical diagnostics related to trace elements in health and disease; and the design of inorganic drugs in chemotherapy, medicine, environmental science, food science, agriculture, toxicology, and biogeology. Other metals in chemistry assist in functional biological sciences. Commonly used techniques include atomic absorption spectrophotometry, inductively coupled plasma mass spectrometry, isotope quantitative analysis, infrared spectroscopy, nuclear magnetic resonance, and X-ray absorption spectroscopy [61].

In recent years, increasing reports on metallomics and metabolic diseases have guided significance for healthy diets. Lindeque et al. (2015) evaluated the protective effects of metallothioneins (MTs) on obesity and high-fat-diet-induced effects, such as insulin resistance in male and female MT-1-, MT-2-, and MT-3-knockout mice [62]. Steinbrenner et al. (2011) summarized the current evidence of interference with selenium compounds and insulin-regulated molecular pathways, verifying that hypernutrient selenium intake and high plasma selenium levels are a potential risk factors for T2D [63]. Roverso et al. (2019) recruited 76 pregnant women from the University Hospital of Padua (Italy), half of whom had GDM. Placental samples were collected from maternal whole blood and umbilical cord blood, and the metal groups were determined via ICP-MS analysis. The results showed that Ca, Cu, Na, and Zn concentrations in the umbilical cord blood of GDM patients were higher than those of the controls, while Fe, K, Mn, P, Rb, S, and Si showed opposite trends [64]. Roverso et al. (2015) used a metallurgical database to analyze GDM and the other types of DM, and found that selenium concentrations were higher in GDM than in the other groups [65]. Liu et al. (2012) developed a joint metabolomics and metallomics method using a hypercholesterolemia rat model, involving proton nuclear magnetic resonance spectroscopy, plasma MS, and metallurgical fingerprinting law. The results revealed that V, Mn, Na, and K could be biomarkers for hypercholesterolemia [66]. The omics-driven study of Lopes et al. was based on time-resolved H-NMR metabolomics to study the results of Roux-en-Y gastric bypass (RYGB) surgery [67]. Cox et al. (2013) studied the role of selenoprotein S (SelS) genetic variation in subclinical CVD risk and mortality in T2D patients [68].

## 4. Microbiomics and DM

Microbial-based human nutritional metabolomics is gut microbiology [69]. The total number of gut microbes in a healthy adult is extremely large, and this microbial layer is known as the “human second genome”. In December 2007, the National Institutes of Health presented the Human Microbiome Program, a microbial genome study of the human digestive tract, mouth, vagina, skin, and nasal passages [70]. The human intestinal microflora begins colonizing from the birth of the host and gradually matures as the host grows to reach a relatively steady state. Gut microbial functions include nutritional and metabolic functions, mucosal barrier function, and immune function. Commonly used research techniques in gut microbiology include DGGE, biochips, and RT-PCR.

Using the intestinal microbiome to analyze changes in human microbial community structure and function under health and disease conditions can improve metabolic disease prevention and treatment. Díaz-Rizzolo et al. (2019) divided 182 prediabetic patients over age 65 into obese and non-obese groups, analyzed the FINDRISK score and biochemical parameters of their intestinal microbiota, and found the importance of intestinal microbes [71]. Larsen et al. (2010) reported that human T2D was associated with changes in the gut microbiota composition and that the gut microbiota could be altered to control metabolic diseases. For example, the proportions of Firmicutes and Clostridia were significantly reduced in the DM group compared with the control group. Similarly, class Betaproteobacteria was highly enriched in the DM group compared to the control and positively correlated with plasma glucose [72]. In addition, a review by Aydin et al. (2018) assessed the contributing role of the gut microbiota in human obesity and T2D [73].

Qiao et al. (2018) reported that the intestinal microbiotas of patients with T2D differed significantly in bacterial composition and diversity from the intestinal microbiotas of healthy subjects [74]. Pasini et al. (2019) conducted a metabolic and anthropometric assessment of 30 clinically stable T2D patients and concluded that DM leads to the overgrowth of the intestinal flora, increased intestinal permeability, and systemic low-grade inflammation. They also found that chronic exercise can reduce excessive intestinal floral growth, intestinal leakage, and systemic inflammation [75]. Ohtsu et al. (2019) found that orally administering *Porphyromonas gingivalis* altered the gut microbiota and aggravated glycemic control in streptozotocin-induced DM mice [76]. Chen et al. (2019) investigated and compared the effects of green tea polyphenols (Polyphenon E (PPE)) and black tea polyphenols (theaflavins (TF)) on gut microbiota and DM development in db/db mice [77]. A review by Whang et al. (2019) provided evidence demonstrating the putative interaction between antidiabetic agents and the gut microbiome and discussed the potential of microbiome modulators to manipulate drugs, microbial interactions, and drug metabolism [78].

## 5. Foodomics and DM

The term “foodomics” was first used in 2007 at networking and academic conferences [79] and refers to the use of omics analysis methods to study the components of complex food systems, such as proteins, peptides, amino acids, carbohydrates, lipids, vitamins, and trace elements. Commonly used methods include genomics, transcriptomics, proteomics, metabolomics, and bioinformatics. Foodomics analysis contributes to explaining the responses of individual genomes to specific dietary compositions; explaining the biochemical, molecular, and cellular mechanisms by which certain active ingredients in food constitute health benefits and adverse effects; determining the role of bioactive food components in key molecular pathways; identifying genes and possible molecular biomarkers from pre-onset to onset; determining the overall role and function of the gut microbiome; conducting unintended effects studies of transgenic crops; studying the application of food microbes as delivery systems; studying food pathogen stress adaptations to the response; ensuring food hygiene, processing and storage; comprehensively evaluating food safety, quality, and traceability; and exploring the molecular basis of biological processes, such as the interaction of crops and pathogens and physical and chemical changes during fruit ripening. The holistic approach to environmental reactions explains the phenomena and determines the biological network [80].

Foodomics involves studying food metabolites to provide a basis for healthy diets. Growing evidence suggests that healthy diets rich in fruits, vegetables, nuts, extra virgin olive oil, and fish are beneficial for preventing and controlling various human diseases and metabolic disorders. This is the Mediterranean diet, which is one of the healthiest existing dietary patterns. Proteomic and metabolomic analyses revealed that Mediterranean diets had clinical implications for metabolic and microvascular activities, cholesterol and fasting blood glucose, and anti-inflammatory and antioxidative effects [81]. Therefore, dietary intervention is beneficial in preventing and treating diseases [82]. Olivas-Aguirre et al. (2016) studied cyanidin-3-*o*-glucoside (Cy3G) metabolites and found that they protected against *Helicobacter pylori* infection, age-related diseases, T2D, CVD, and metabolic syndrome [83]. Janšáková et al. (2019) used rat models to verify advanced glycation end products (AGEs) in hot-processed foods believed to cause GDM and found that these AGEs did not cause disease [84]. Takahashi et al. (2015) performed comprehensive proteomic and metabolomic analyses verifying that coffee consumption leads to increased ATP conversion and demonstrating that coffee consumption can help prevent DM [85]. Inulin has been reported to possess a significant number of diverse pharmaceutical and food applications. A review by Tsurumaki et al. (2015) described the current status of utilizing omics technologies in elucidating the impact of inulin and inulin-containing prebiotics at the transcriptome, proteome, metabolome, and gut microbiome levels to fully illustrate the intricate beauty behind the relatively modest influence of food factors, like inulin, on host health. [86]. Alkhatib et al. (2017) reported that functional foods contain biologically active ingredients associated with physiological health benefits and preventing and controlling chronic diseases, such as T2D. Zhao C et al. (2019) made a systematic report on glucose metabolism in T2D as well as to explore the relationships between natural phytochemicals and glucose handling [87]. Sébédio et al. (2017) proposed metabolomics studies to discover new biomarkers of early metabolic dysfunction and to predict biomarkers for developmental pathology (e.g., obesity, metabolic syndrome, and T2D) but focused on developing methods to identify and validate biomarkers for nutrient exposure [88].

## 6. Conclusions

Nutrition is the study of beneficial food ingredients and the science of human intake and use of these ingredients for health. The main purpose of studying nutrition is to prevent diseases (especially metabolic diseases) and protect human health through a reasonable daily diet. With the development of omics technology, nutritional development has flourished, and the term “precision medicine” has been proposed. Precision medicine is an emerging approach to disease prevention and treatment that considers differences in personal genetics, environment, and lifestyle. The concept of precision medicine is becoming increasingly popular. Using large amounts of data, genomics, and other “omics”, such as metabolomics, proteomics, and transcriptomics, can make personalized medicine a reality in the near future [89]. In addition, the Metabolomics Association believes that integrating metabolomic data into precision medicine programs is timely and will provide extremely valuable new data to complement the current data [90]. Figure 1 shows the branching method of human nutrition on DM research. Figure 2 shows the application of human nutrition on DM research. Table 1 lists the results of recent studies on preventing and treating DM according to the branches of nutriomics, specifically the factors that are highly relevant to DM, such as metal elements, SNPs, and pathways. This review summarized the results obtained by various researchers using analytical omics methods and provides a reference for subsequent research.

## Figures and Tables

**Figure 1 ijms-20-05375-f001:**
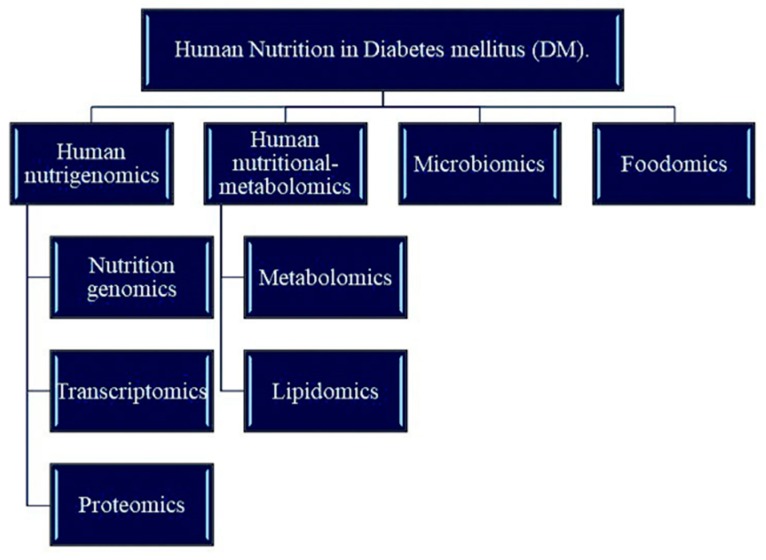
Advances in research on human nutrition in diabetes mellitus (DM).

**Figure 2 ijms-20-05375-f002:**
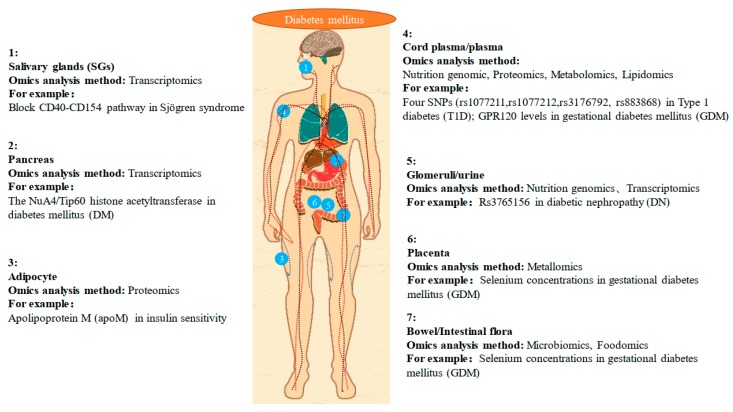
The application of human nutrition on diabetes mellitus (DM) research.

**Table 1 ijms-20-05375-t001:** Advances in research on human nutrition in diabetes mellitus (DM).

Title	Branch	Application Example	References
Human nutrigenomics	Nutrition genomics	Three loci associated with T2D were identified in the non-coding regions near CDKN2A and CDKN2B, introns of IGF2BP2 and CDKAL1 introns, and replication associations near HHEX and SLC30A8;	Defesche et al., 2017 [18]
		Rs3765156 in PIK3C2B was significantly associated diabetic nephropathy (DN);	Jeong et al., 2019 [20]
		Four SNPs (rs1077211,rs1077212,rs3176792, rs883868) could alter enhancer, H3K4me1 and H3K27ac, activity in T1D;	Gao et al., 2019 [21]
	Transcriptomics	Block CD40-CD154 pathway interaction can inhibit ectopic lymphoid structures and Sjögren syndrome;	Wieczorek et al., 2019 [23]
		The cohesion loading complex and the NuA4/Tip60 histone acetyltransferase complex play a key role in regulating insulin transcription and release;	Fang et al., 2019 [26]
		Transcriptome analysis of glomerular endothelial cells in DM mice revealed up-regulated leucine-rich α-2-glycoprotein 1 (LRG1);	Hong et al., 2019 [27]
		The transcriptomic data of post-mortem Alzheimer’s disease (AD) and T2D brains revealed the main role of autophagy in the molecular basis of AD and T2D;	Caberlotto et al., 2019 [28]
		Islet cell transcriptome data from control and DM mice revealed three new target genes (*Upk3a*, *Adcy1*, and *Dpp6*) differentially expressed genes in the transcriptome of pancreatic alpha cells;	Dusaulcy et al., 2019 [29]
	Proteomics	Apolipoprotein M (apoM) may be associated with insulin sensitivity;	Sramkova et al., 2019 [32]
		Many immunologically related proteins, including heparin cofactor 2, Ig α-1 chain C region, zinc-α-2-glycoprotein, are differentially expressed in T2D;	Abdulwahab et al., 2019 [33]
		The site-specific glycation of red blood cell proteome was identified with different glycemic index in diabetic patients by using the nanoLC/ESI-MS proteomics platform;	Muralidharan et al., 2019 [34]
		Used sequential window acquisition of all theoretical fragment ion spectroscopy (SWATH) mass spectrometry (MS) to find that hemoglobin A1c (HbA1C) levels decrease with weight loss and insulin sensitivity improve;	Malipatil et al., 2019 [36]
Human nutritional-metabolomics	Metabolomics	Five amino acids (tyrosine, alanine, isoleucine, aspartic acid, and glutamic acid) were found to be significantly associated with an increased risk of developing T2D;	Vangipurapu et al., 2019 [46]
		CANA regulates key nutrient sensing pathways, activates 5’AMP-activated protein kinase (AMPK), and inhibits rapamycin (mTOR) independent on insulin or glucagon sensitivity or signaling;	Osataphan et al., 2019 [48]
		GPR120 levels were associated with GDM;	He et al., 2019 [51]
	Lipidomics	Sphingomyelin was lower in T1D progression;	Lamichhane et al., 2018 [52]
		FFA C16:0 and 16:0-LPA lipids may be potential candidates for the diagnosis and study of obesity-related diseases;	Wang et al., 2019 [53]
		It has been discovered that the ratios of C=C isomers were much less affected by interpersonal variations than their individual abundances, suggesting that isomer ratios may be used for the discovery of lipid biomarkers, which can also be used for subsequent predictive screening for DM;	Zhang et al., 2019 [54]
		Lipidomics analysis found that *Cyclocarya paliurus* (CP) may be the cause of diabetic dyslipidemia;	Zhai et al., 2018 [58]
		Supranutritional selenium intake and high plasma selenium levels are potential risk factors for T2D;	Steinbrenner et al., 2011 [63]
	Metallomics	The concentrations of Ca, Cu, Na, and Zn in the umbilical cord blood of GDM were higher than those of the control samples, while Fe, K, Mn, P, Rb, S, and Si showed opposite trends;	Roverso et al., 2019 [64]
		Selenium concentrations in GDM were higher than others;	Roverso et al., 2015 [65]
		V, Mn, Na, and K may be biomarkers for hypercholesterolemia diseases;	Liu et al., 2012 [66]
Microbiomics		Human T2D is associated with changes in the composition of the gut microbiota, for example, the proportions of phylum Firmicutes and class Clostridia were significantly reduced in the DM group compared to the control group;	Qiao et al., 2018 [74]
		T2D intestinal microbiota was significantly different from the intestinal microbiota of healthy subjects. It has been confirmed that using the fermentation products of *Paenibacillus bovis* sp. nov. BD3526 to treat the Goto-Kakisaki (GK) rats can improved its related symptoms;	Ohtsu et al., 2019 [76]
		Oral administration of *Porphyromonas gingivalis* altered the gut microbiota and aggravated glycemic control in streptozotocin-induced DM mice;	Olivas-Aguirre et al., 2016 [83]
Foodomics		Metabolites of cyanidin-3-*O*-glucoside (Cy3G) and found it to protect against *Helicobacter pylori* infection, age-related diseases, T2D, cardiovascular disease, and metabolic syndrome.	Alkhatib et al., 2017 [81]

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
