# Peer review of "Advances in Research on Diabetes by Human Nutriomics"

_ijms, 2019, doi:10.3390/ijms20215375_

Round 1

Reviewer 1 Report

Reviewed article "Advances in research on diabetes by human nutriimics" is very interesting as well as important. However, there are few comments/suggestions/errors.

Lines 24 and 26. There are "Type I diabetes" and "Type II diabetes". It is incorrect. It should be "Type 1 diabetes" and "Type 2 diabetes" Line 35. It is "... with 11,4294.8 DM patients." Which value is 11,4294.8 Many articles cited in text need changes. For example: line 73 is "Jeong et al." It should be "Jeong et al. [2019], line 88. It should be "Wieczorek et al. [2019] and many times in text. Line 103. Alzheimer's diseases is also suggested as Type 3 Diabetes. Line 120. It is "... which is lower..." What is lower? Level/, activity? Lines 140,141. It is "Seven ..... diagnose DM" There are described (in details) laboratory tests to diagnose DM. Therefore this sentence should be treated as suggestion, additional tests to diagnose DM, not as main markers. Line 163. It is "... of Imperial"... > "... from Imperial..."? Line 176 There are "sphagnum, konjac and aloe vera .." It should be "Sphagnum, Konjac and Aloe vera" See also line 177 Line 214. "... PAS kinases" Write please full name "PAS" Lines 222-223. "Zhai et al...... diabetic dyslipidemia" This sentence needs additional sentences as explanation. Lines 267-274. This problem "human gut microbiota-type 2 DM" is very important, however is also difficult and controversial. There are several changes in composition of gut microbiota, not only proportions Firmicutes and Clostrida. Literature on this subject is very rich. Therefore, write please more on this dependence.  Lines 303-305. This is Mediterranean diet, which is one of the healthiest dietary pattern existing. Lines 310-315. What is cause, what is effect? References need changes. Compare please, for example No. 23 and 24. Reference 23 is correct (full details), reference 24 - only year. 

Author Response

Point1: Lines 24 and 26. There are "Type I diabetes" and "Type II diabetes". It is incorrect. It should be "Type 1 diabetes" and "Type 2 diabetes"

Point2: Line 35. It is "... with 11,4294.8 DM patients." Which value is 11,4294.8

Ponit3: Many articles cited in text need changes. For example: line 73 is "Jeong et al." It should be "Jeong et al. [2019], line 88. It should be "Wieczorek et al. [2019] and many times in text.

Ponit4: Line 103. Alzheimer's diseases is also suggested as Type 3 Diabetes.

Thanks for pointing out that these points have been revised in the manuscript.

Point5: Line 120. It is "... which is lower..." What is lower? Level/, activity?

The revised content is as follows: Sramkova et al. [2019] identified apolipoprotein M (apoM) by transcriptome and proteomic analysis of conditioned media from human adipose tissue (AT)-isolated adipocytes and stromal cells, which concentrations is lower in subjects with metabolic syndrome and T2D and may be associated with insulin sensitivity

Point6: Lines 140,141. It is "Seven ..... diagnose DM" There are described (in details) laboratory tests to diagnose DM. Therefore this sentence should be treated as suggestion, additional tests to diagnose DM, not as main markers.

The revised content is as follows: Seven upregulated proteins and 4 downregulated proteins in DM patients can be used as additional tests to diagnose DM.

Point7: Line 163. It is "... of Imperial"... > "... from Imperial..."?

Point8: Line 176 There are "sphagnum, konjac and aloe vera .." It should be "Sphagnum, Konjac and Aloe vera" See also line 177

Point9: Line 214. "... PAS kinases" Write please full name "PAS"

Thanks for pointing out that these points have been revised in the manuscript.

The revised content is as follows: Pape et al. [2018] explored the role of high-fat-diet-induced hepatic triglycerides in high-fat-diet-induced DM by MS-based lipidomics and found correlations between Per-Arnt-Sim Kinase (PASK)

Point10: Lines 222-223. "Zhai et al...... diabetic dyslipidemia" This sentence needs additional sentences as explanation.

The revised content is as follows: Zhai et al. [2018] used lipidomics analysis to find that Cyclocarya paliurus may improve diabetic dyslipidemia by reducing accumulation of hepatic lipid droplets and regulating circulatory lipids in diabetic mice on PI3K signaling and MAPK signaling pathways.

Point11: Lines 267-274. This problem "human gut microbiota-type 2 DM" is very important, however is also difficult and controversial. There are several changes in composition of gut microbiota, not only proportions Firmicutes and Clostrida. Literature on this subject is very rich. Therefore, write please more on this dependence. 

The revised content is as follows: Larsen et al. [2010] reported that human T2D was associated with changes in the gut microbiota composition and that the gut microbiotas could be altered to control metabolic diseases. For example, the proportions of Firmicutes and Clostridia were significantly reduced in the DM group compared with the control group. Similarly, class Betaproteobacteria was highly enriched in DM compared to control and positively correlated with plasma glucose [74]. In addition, a review by Aydin et al. [2018] assessed the contributing role of the gut microbiota in human obesity and type 2 diabetes (T2D) [75].

Point12: Lines 303-305. This is Mediterranean diet, which is one of the healthiest dietary pattern existing.

Point13: Lines 310-315. What is cause, what is effect?

This section focuses on the use of Foodomics analysis to find out the impact of diet on metabolic diseases. Many literatures have reported relevant research progress. This paper clarifies the importance of foodomics and gives examples for reference. Therefore, this part is just an example not to distinguish cause or effect.

Point 14: References need changes. Compare please, for example No. 23 and 24. Reference 23 is correct (full details), reference 24 - only year. 

Thanks for pointing out that the point have been revised in the manuscript.

Reviewer 2 Report

The manuscript deals with an interesting and important medical and nutritional problem. THe article is relevant, interesting and informative. I suggest only few minor changes

Line 34 “In 2017, China ranked first worldwide with 11,4294.8 DM patients.” The number is not correctly written?

Line 307 “cyanidin-3-O-glucoside“ O should be italic

Line 175: “Chen et al. found through metabolomics analysis that glucomannan in sphagnum, konjac and aloe vera leaves can balance disorders of the urea cycle, lipid metabolism, glucose and amino acids. Specifically, konjac glucomannan treatment is more effective in regulating lipids and glucose [47].” In the way this sentence is written, its inclusion in the metabolomic anaylsis is not justified. In the present way, it just represent therapeutic approach using phytochemicals. Thus, the sentence should be rewritten in such a way that the metaboolomics contribution is more specific and visible. Alternatively, it may be omitted.  

Author Response

Point1: Line 34 “In 2017, China ranked first worldwide with 11,4294.8 DM patients.” The number is not correctly written?

According to the International Diabetes Federation (IDF), China ranked first worldwide in 2017 , which value is 11,4294.8. In the top 10 countries with the largest number of diabetic patients in 2017, the top three were China, India and the United States, and the number of diabetic patients (20-79 years old) was 114.4 million, 72.9 million and 30.2 million, respectively. It is estimated that by 2045, the top three countries with the largest number of diabetic patients will be India, China and the United States, with a total of 134.3 million, 119.8 million and 35.6 million respectively.

Point2: Line 307 “cyanidin-3-O-glucoside“ O should be italic

Thanks for pointing out that the point have been revised in the manuscript.

Point3: Line 175: “Chen et al. found through metabolomics analysis that glucomannan in sphagnum, konjac and aloe vera leaves can balance disorders of the urea cycle, lipid metabolism, glucose and amino acids. Specifically, konjac glucomannan treatment is more effective in regulating lipids and glucose [47].” In the way this sentence is written, its inclusion in the metabolomic anaylsis is not justified. In the present way, it just represent therapeutic approach using phytochemicals. Thus, the sentence should be rewritten in such a way that the metaboolomics contribution is more specific and visible. Alternatively, it may be omitted.

The revised content is as follows: Chen et al. [2019] used metabolomics to analyze rat blood and urine samples treated with Sphagnum, Konjac and Aloe vera leaves-derived glucomannan. It was found that glucomannan balances urea cycle, lipid metabolism, glucose and amino acid disorders. And we think it has reference value which suggests that metabolomic analysis is a common experimental method.Do you think so?